# Tissue Adhesives in Reconstructive and Aesthetic Surgery—Application of Silk Fibroin-Based Biomaterials

**DOI:** 10.3390/ijms23147687

**Published:** 2022-07-12

**Authors:** Ralf Smeets, Nathalie Tauer, Tobias Vollkommer, Martin Gosau, Anders Henningsen, Philip Hartjen, Leonie Früh, Thomas Beikler, Ewa K. Stürmer, Rico Rutkowski, Audrey Laure Céline Grust, Sandra Fuest, Robert Gaudin, Farzaneh Aavani

**Affiliations:** 1Department of Oral and Maxillofacial Surgery, Division of Regenerative Orofacial Medicine, University Hospital Hamburg-Eppendorf, 20251 Hamburg, Germany; nathalie.peter@stud.uke.uni-hamburg.de (N.T.); a.henningsen@uke.de (A.H.); leonie.frueh@t-online.de (L.F.); s.fuest@uke.de (S.F.); f.aavani@uke.de (F.A.); 2Department of Oral and Maxillofacial Surgery, University Medical Center Hamburg-Eppendorf, 20251 Hamburg, Germany; t.vollkommer@uke.de (T.V.); m.gosau@uke.de (M.G.); p.hartjen@uke.de (P.H.); r.rutkowski@uke.de (R.R.); a.grust@uke.de (A.L.C.G.); 3Department of Periodontics, Preventive and Restorative Dentistry, University Medical Center Hamburg-Eppendorf, 20251 Hamburg, Germany; t.beikler@uke.de; 4Department of Vascular Medicine, University Heart Clinic Translational Wound Research, University Medical Center Hamburg-Eppendorf, 20251 Hamburg, Germany; e.stuermer@uke.de; 5Department of Oral and Maxillofacial Surgery, Charité-Universitätsmedizin Berlin, 12200 Berlin, Germany; robert-andre.gaudin@charite.de

**Keywords:** wound healing, tissue adhesives, silk fibroin, silk-based adhesives, biopolymers

## Abstract

Tissue adhesives have been successfully used in various kind of surgeries such as oral and maxillofacial surgery for some time. They serve as a substitute for suturing of tissues and shorten treatment time. Besides synthetic-based adhesives, a number of biological-based formulations are finding their way into research and clinical application. In natural adhesives, proteins play a crucial role, mediating adhesion and cohesion at the same time. Silk fibroin, as a natural biomaterial, represents an interesting alternative to conventional medical adhesives. Here, the most commonly used bioadhesives as well as the potential of silk fibroin as natural adhesives will be discussed.

## 1. Introduction

The earliest records of surgical wound care date back to 1100 BC. At that time, ligatures made of leather were used to close abdominal wounds [1]. Over the years, suture materials have evolved, but the basic principle of wound closure remains unchanged: adaptation of wound edges with minimal tissue tension. Ideally, the method should be simple, safe to use, quick, inexpensive, as painless and bactericidal as possible, and should produce an optimal cosmetic result [2]. However, a procedure that optimally meets all of these requirements does not yet exist.

Currently, there are various methods of wound care that are used in everyday clinical practice. These include sutures, staplers, or wound closure tapes, which have a wide variety of advantages and disadvantages. Alternatively, tissue adhesives can be used to attach skin, mucosa, or muscle, allowing the natural healing process to take place [3].

Currently, suturing is considered the most basic and common method of tissue union in all surgical disciplines. Synthetic and biological sutures as well as absorbable and non-absorbable sutures are used in this process. Appropriate tensile strength and low dehiscence are important here [2,4]. In addition, the penetration of healthy tissue creates a possible entry portal for pathogens, thus increasing the risk of infection [5]. Non-absorbable material has to be subsequently removed, requiring another uncomfortable and costly intervention. This method is associated with further complications, such as nerve lesion, postoperative adhesions, and necrosis due to injury to blood vessels [2].

So-called stackers represent an alternative. The application is faster and simple. However, healthy tissue must also be injured here; thus, inflammation and significant scarring are more likely. In addition, the failure rate of the procedure is relatively high. For complex, uneven lesions, stackers are also rather unsuitable. Closure strips or tapes are another option since their quick and easy application and the lack of risk of needlestick injuries leads to their common use in clinical settings. The material is inexpensive, has a low risk of infection, and does not require professional removal. The cosmetic results are also convincing. However, tapes have the lowest tensile strength, and thus, they have the greatest risk of dehiscence. This also limits their use to linear, tension-free, dry wounds without body hair [2,5].

Another problem that has been sonographically shown to occur in variable degrees in 100% of postoperative patients using the above-mentioned methods is the formation of a seroma. This is an accumulation of lymphatic fluid in the dead space between the wound edges, which is regularly treated with the aid of drains and aspiration [6]. For the patient, these procedures mean further pain, increased risk of foreign body infection, prolonged healing time, and prolonged hospitalization. The development of a tissue adhesive that specifically adheres to moist surfaces, minimizing free space within the tissue could reduce seroma formation, and thus, revolutionize the surgical standard of care [6].

Numerous chemical and mechanical wound closure materials have been developed over the past decades [5]. The most common methods of wound care in reconstructive and aesthetic surgery and the current state of the art of tissue adhesives are presented below. In order to highlight silk protein and its potential as well as research gaps, this review focuses specifically on the development and use of silk-based tissue adhesives.

## 2. Principals of Tissue Adhesives

According to their function, tissue glues can be categorized into hemostats, sealants, and adhesives. Correspondingly, a hemostat is a substance that provokes blood clots and ceases to function without blood. A sealant also establishes a barrier layer to prevent fluid or gas escape, and an adhesive bond keeps the two surfaces together effectively through different mechanisms [7,8].

Generally, tissue adhesives are being placed between tissues to hold the edges of the wound together [9]. The function of adhesives, for the purpose of providing adhesion between two surfaces, relies on a combination of adhesive and cohesive strength. Accordingly, for adhesive strength, there must be strong intermolecular forces maintaining the bond between the adhesives and the adherent tissue surface, and for cohesive strength, there must be tight internal forces of the adhesive for holding the network together [10].

Therefore, it is essential to find the right balance between these two parameters for owing a satisfactory adhesiveness [11]. Mechanical interlocking, intermolecular bonding, electrostatic bonding, chain entanglement, and cross-link creation are examples of adhesive and cohesive interactions.

An applicable tissue adhesive should be safe to use for both patient and practitioner. Thus, the agent must have neither toxic nor hemolytic properties and elicit only a minimal immune response (biocompatibility). In addition, any carcinogenic effect must be prevented [12].

Natural wound healing should be promoted but in no case disturbed and could even be accelerated by integrated drugs or growth factors [5]. The aforementioned requirements can be tested both in vitro using cell cultures and in animal experiments related to postoperative infection rates [2]. In addition, the tissue adhesive must be safe and effective in its action and for the surgical indication in question [5]. Both experimentally and clinically based studies are suitable for this purpose, with the majority choosing to compare new materials to commercially available wound closure methods in terms of wound dehiscence and time to closure for incisions of variable length and complexity [2]. Deformation behavior, tensile strength, and adhesiveness can be studied using physical testing methods [13,14]. In addition, the curing, bonding, and degradation time of the adhesive must be modifiable for different applications. Biodegradation can only take place once the wound has healed sufficiently (ideally starting after about three weeks, with complete degradation after three months [5]). In conclusion, the swelling index (swelling value), i.e., the retention capacity for water, must be minimal to prevent compression of surrounding structures [15]. The compound should also adhere securely to moist surfaces [12], which would enable internal application and avoid the development of a postoperative seroma. Effectiveness in this regard can be determined sonographically [6].

On the other hand, there are some tissue considerations regarding developing an ideal tissue adhesive. The tissue characteristics can be classified as mechanical properties, surface characteristics, and local environment. Every tissue exhibits specific mechanical properties, such as elasticity, stiffness, and rigidity. The satisfactory performance of tissue adhesives depends intensely on their ability to match the chemical and physical properties of the underlying tissue for the required period of time [16,17]. Tissue microarchitecture, regardless of the type of the adhesive interactions, is a significant factor determining the potential of polymers to penetrate and interlock within the tissue [18,19]. The tissue macro and microenvironment will also adjust the material performance over time. For example, pH variations, oxidative species, or endogenous enzymes can directly affect the material characteristics by reducing both the adhesive and cohesive strengths. One of the more important factors to develop an ideal tissue adhesive is tissue regeneration time. This factor alters the stability of the material while ensuring the maintenance of material adhesive properties. In addition, the dynamic behavior of the fabric emphasizes the need for fatigue resistance of the material. Flexibility of tissue adhesives is another critical requirement for applications where the adhesive material must be bent or twisted (e.g., topical sealants in the neck or knee). Elasticity is also critical for applications such as lung sealants, where the size of the underlying tissue changes significantly over time [20]. Table 1 displays some specific characteristics of adhesives regarding tissue types.

The first adhesive substances were synthesized in the form of cyanoacrylates (CA) by a German chemist named Ardis, as early as 1949, and are currently attracting increasing attention [21]. However, these substances are not free of undesirable side effects, such as minor skin irritation or allergic reactions [22]. A relatively new approach is the development of biomimetic adhesives. The focus lies on the transfer of phenomena from nature to technology. The best-known representatives of bioadhesion are tissue adhesives based on mussel proteins. In addition, adhesives based on gecko proteins, indoparasites, and worms have already been investigated [5,12].

Currently, the Department of Oral and Maxillofacial Surgery at the University Medical Center Hamburg-Eppendorf is developing such a biomimetic adhesive based on caterpillar silk with an industrial partner. Silk proteins show promising possibilities to modify chemical and mechanical properties of biomedical products [23].

Silk can be processed into a wide variety of forms, such as gels, coatings, nanofibers, membranes, or scaffolds, or combined with other substances and drugs, making it an attractive option for numerous application areas [23]. With the help of improved understanding of the interaction between silk and living cells, adhesives with extremely high strength could be produced through structural modifications. In addition, caterpillar silk could be an ideal basis for a tissue adhesive through customization of its absorbability, antibacterial properties, and potentially self-healing properties [6,23].

## 3. Categories of Tissue Adhesives

### 3.1. Synthetic Polymers

Polycyanoacrylates have been developed since 1949 and first found their medical application as tissue adhesives ten years later by Cooler et al. [5]. Polycyanoacrylate tissue adhesives demonstrate antimicrobial properties [24], showing an activity against gram-positive microorganisms such as *S. aureus* and *S. pneumoniae*, thus destroying their cell walls during polymerization [25,26].

The liquid monomer polymerizes within seconds at room temperature without the addition of a catalyst [5]. Adhesion occurs via covalent bonds between the cyanoacrylates and the functional groups of the tissue proteins. Due to its electron-repelling nitrile group, the acrylic compound is susceptible to nucleophilic attack by weak bases such as water or amines [5]. In an exothermic reaction, this creates a stable cross-link with the skin [4].

Some characteristics of the Polycyanoacrylates compounds can be directly affected by their carbon side chains. For instance, the longer carbon side chains provide more flexibility and stability for these compounds [2,4].

Initially, short-chain derivatives (methyl and ethyl 2-cyanoacrylates) were developed. However, the rapid degradation and accumulation led to inflammatory reactions through the release of histotoxic formaldehyde and cyanoacetate [5].

In Europe and Canada, n-butyl 2-cyanoacrylates in particular found their use [5] (However, this material has an extremely low fracture strength and is particularly brittle [2,5]. In contrast, the development of long-chain CAs (octyl-2 cyanoacrylate, Dermabond^®^, Ethicon Inc., Somerville, New Jersey) proved far more successful [27]. The substance is degraded after approximately 7–10 days before histotoxic degradation can even occur. Thus, the release of formaldehyde and cyanoacetate no longer plays a role, so that the product was also approved in the USA by the FDA (Food and Drug Administration) in 1998 [5]. In addition, improved flexibility and tensile strength of the material could be achieved [2].

Tissue adhesives made from cyanoacrylates are currently the most widely used. Due to initial concerns caused by the toxicity of short-chain CAs, acceptance has only been steadily increasing in recent years in many surgical disciplines [5].

Another approach considers at synthetic sealants made from polyethylene glycol (PEG). These PEG sealants form hydrogels designed to seal tissues from fluid leakage. Substances already in use show acceptable biocompatibility and are hardly recognized by the immune system due to a kind of stealth effect. The material is water-soluble and binds even to moist surfaces [28]. Sufficient elasticity and ductility are also among their advantages [4]. However, a major safety risk is the particularly high swelling index, whereby PEG-based materials can swell up to 400% of their original size, compressing nerves and vessels in the immediate vicinity and clogging application materials [5]. Furthermore, the application is extremely complicated. The product consists of two components, which must be stored differently and applied very quickly by the practitioner via a syringe due to the short drying time. Cohesion is rather poor and the substance quickly becomes brittle. In addition, these products are very expensive. Due to their numerous undesirable side effects, these adhesives are not yet suitable to replace sutures entirely [5].

Polyurethane (PU) can be employed for different tissue adhesive or sealant purposes. This synthetic polymer represents excellent thermal stability at physiological temperature and is also being applied for bone fixation, hemostasis, and sealing of vascular grafts in several surgery procedures [29,30]. The most recognized commercial polyurethane-based adhesive is TissuGlu^®^ Surgical Adhesive, which is biodegradable and used for abdominal tissue bonding. This adhesive consists of a hyperbranched polymer with isocyanate end groups containing about 50 wt.% of lysine. A frequently experienced side effect after using this adhesive for abdominal or breast surgery is fluid accumulation under the skin, resulting in a so-called seroma [31]. 

Aliphatic polyesters, such as poly(Ɛ-caprolactone) (PCL) and poly(lactic-co-glycolic acid) (PLGA), have also been applied as tissue adhesives. A recently commercialized example of a tissue adhesive based on PLGA is TissuePatch^TM^. This tissue adhesive and sealant is applied in the prevention of air leakage after lung surgery or avoiding fluid leakage after surgery on soft tissue [32].

Dendrimers are a type of synthetic, fully branched polymers with a central core. The addition of a branching layer develops different types containing large number of functional groups around their perimeter. Because of this unique structure, dendrimers are appropriate cross-linkable building blocks for tissue adhesives [33]. The first dendritic tissue adhesive was reported for ophthalmology in 2002 [34]. These dendritic structures were fabricated by a poly(glycerol succinic acid) dendrimer (PGLSA) and a PEG linear polymer. It was suggested that the developed adhesives are as effective as traditional sutures in closing a corneal incision in in vitro studies.

Other synthetic tissue adhesives are mainly considered in a scientific context and have not yet been able to establish themselves in clinical settings. However, several synthetic tissue adhesives are commercially available by now.

### 3.2. Polysaccharide-Based Adhesives

Polysaccharide-based tissue adhesives have the advantage that their natural occurrence makes it easier to develop materials that are biodegradable, biocompatible, and less immunogenic. Chitin and chitosan are present in the exoskeleton of invertebrates and in the cell wall of fungi and exhibit the above-mentioned properties. It has been reported that chitin and chitosan gel-based substances have inhibitory effects on tumor angiogenesis and metastasis [35,36], and by inducing apoptosis, can prevent tumor cell proliferation [36]. In addition, they potentially have antimicrobial activity and attract red blood cells due to their positive charge, thus accelerating local coagulation [37]. Research to date has focused particularly on the development of wound dressings, tissue engineering, and drug delivery systems [38]. Commercially, biopolymers are produced from shrimp shell residues. The process is particularly complex and the extraction is costly [39]. Due to their dense crystal structure, previous products are extremely poorly soluble. Alternative wound adhesives based on polysaccharides, such as dextran, chondroitin sulfate, and hyaluronic acid, are also being investigated in studies. At the current time, no polysaccharide-based tissue adhesives are commercially available.

### 3.3. Protein-Based Adhesives

As the most common protein-based representatives, fibrin glues are biocompatible due to their natural ingredients, degrade within a few days to weeks depending on their composition, and can, therefore, be used both locally and internally. Since these products are considered to be relatively safe in terms of their biocompatibility, infectivity and natural degradation, they are used in numerous specialist disciplines. In vascular surgery, they are now regularly used to control unstoppable bleeding [5]. As a sealant, targeted air- and fluid-tight closure can be achieved in lung and neurosurgical procedures [40]. In plastic and aesthetic surgery, bleeding from burn wounds can be treated after debridement and flapoplasties can be adapted [41]. Furthermore, liver and spleen injuries have been successfully treated in visceral surgery with tissue adhesives [42]. Fibrin is a physiological component of human blood and an essential factor in secondary hemostasis [12]. Products have been commercially available in Europe since 1972 [43,44]. Adhesives consist of two separate components: Thrombin (Factor IIa) and Calcium (Factor IV) as well as Factor VIII and Fibrinogen [12].

First, cleavage of fibrinogen by factor IIa to fibrin monomers and subsequent cross-linking via covalent bonds of lysine and glutamine residues by factor VIIIa and calcium form an insoluble thrombus. This bond is further stabilized by natural fibrinolysis inhibitors (a2-P inhibitor, a2 macroglobulin, plasminogen activator inhibitor 2) [12]. However, the use of fibrin glues is relatively complicated because fibrinogen and thrombin must be kept separately and refrigerated. Prior to application, the components must then be warmed and dissolved before transferring them to a dual-chamber syringe. Consequently, the preparation time is relatively long [5]. Since human blood products are involved, contamination with infectious agents, such as HIV or hepatitis, cannot be completely excluded. Alternatively, material can be derived from porcine blood, but this may lead to allergic or autoimmune reactions. Compared to synthetic Polycyanoacrylates, the adhesion ability is significantly inferior, especially under humid conditions [5].

Gelatin and albumin are two naturally occurring proteins and can be used in combination with other components as tissue adhesives. The best-known product is gelatin-resorcinol-formaldehyde-glutaraldehyde, GRFG (BioGlue, CryoLife Inc., Kennesaw, GA, USA) [5]. Under dry conditions, adhesion strength is significantly superior to fibrin adhesives and comparable to cyanoacrylates, but decreases significantly in humid environments [12]. The cytotoxic properties of formaldehyde and glutaraldehyde can cause inflammation and increase scarring, which is why further research is needed to investigate alternative drug combinations.

### 3.4. Biomimetic Adhesives

Artificially produced adhesives have not yet come close to substances found in nature in terms of properties such as biocompatibility. Our knowledge of natural adhesion systems is far from producing innovative tissue adhesives for commercial use [45]. Therefore, it is essential to explore the individual composition of these materials and understand their mode of action and function in order to transfer them to biomedical use [45,46]. From an evolutional perspective, animals have been able to develop multiple mechanisms to walk, climb, or adhere to different surfaces [45]. This phenomenon is being exploited in bioadhesive research. However, there are still difficulties in developing tissue adhesives that form sufficiently stable bonds even under humid conditions [47]. Therefore, great efforts have been made in the past two decades to explore synthetic adhesives based on mussel proteins [12,46].

Some mussel species are capable of adhering securely to a wide variety of surfaces under extreme conditions. By secreting so-called byssus, a protein-rich secretion produced in the animals’ foot glands, they can survive in waters with strong currents, salty environments, and pH and temperature fluctuations with ease. These mussel foot proteins (MFPs) are rich in catecholamines such as L-3,4-dihydroxyphenylalanine (DOPA), and form fibers that connect the mussel to a contact surface via electrostatic interactions, hydrogen bonds, and covalent cross-links [12,47]. Moreover, it is indicated that oxidation of DOPA by metal ions or enzymes is mandatory to adhere to surfaces or tissue [48].

Numerous attempts have been made to extract these mussel proteins. One challenge is that several thousand mussels are needed to extract just one gram of protein, which is why one has to rely on synthetic production [47]. The artificial mussel proteins show little antigenic activity and function well under dry conditions [12]. However, in humid environments or water, the binding strength strongly decreases. This is because catecholamine groups are particularly responsive to oxidation under neutral and alkaline conditions, which weakens the adhesion ability [46,47]. Furthermore, the long drying and degradation time of some adhesives is another drawback, which has limited clinical applications to date. Therefore, nowadays, biomaterial researchers have been trying to fabricate biomimetic synthetic polymers comprising DOPA moieties [49,50].

Yin et al. fabricated a DOPA-modified silk fibroin-based bioadhesive and chemically cross linked the structure using genipin [51]. Furthermore, metal ions have also been utilized to modify the adhesion properties of adhesive (dopamine modified). The results demonstrated that the DOPA-modified silk fibroin-based composite shows a greater stickiness except slow gelation speed. In addition, they proved the doping of cationic metal ions can hasten the gelation of the bioadhesive.

Other examples of biomimetics include adhesives based on gecko proteins, endoparasites, and worms. These also show good biocompatible properties, but again, the extraction of the substances is problematic and the adhesion strength in water is limited [45].

The primary concerns of applying mussel adhesive protein mimics are their long degradation times and the utilization of harmful oxidizing agents such as periodate and iron (III). Although DOPA-functionalized polymers show high potential as tissue adhesives, no clinical studies have been performed to date [34].

## 4. Evaluation Methods of Tissue Adhesives

After developing a surgical tissue adhesive, some standard testing methods should be taken into account to evaluate its different characteristics and performance. Because each tissue has unique physical and biological properties, focusing on a specific evaluating method cannot be a good idea [11]. Therefore, it is necessary to develop in vitro or ex vivo models very similar to in vivo conditions for performance analysis in different tissues. Additionally, to perform a better evaluation, mimicking the biological state should be considered. For instance, the developed samples can be tested at the biological temperature or exposed to similar physiological forces that they will experience later in the tissue.

When the bioadhesive is applied in the body, it will interact with its surrounding elements and be affected through time. Hence, examining the adhesive properties over time at physiological conditions is a critical factor that should be considered.

On the other hand, the characteristics of the tissue adhesive should be maintained for an extended duration. Depending on the targeted tissue and application, this time may vary from days to months. With regards to adhesion measurement, both direct and indirect adhesion should be assessed. Direct adhesion measurements are commonly used to achieve qualitative or quantitative adhesive characteristics data. Particular techniques, such as tension tests, peel tests, lap and shear tests, and pull-off /pull-out tests, are classified as direct adhesion measurement [52]. Tension testing is widely used to measure the strength of adhesives. The tensile test results provide a precise profile of the force, extension, and time, including partial failure, slippage, sharpness, and percentage of breaking. For adhesive testing, the data curve of the force vs. extension is employed to gain ultimate tensile strength [53].

Peel tests are commonly accomplished to evaluate the strength of bonded connections during exposure to peel force [54]. In this measurement approach, the speed and angle of separation are evaluated. Because these methods are extremely sensitive to the smallest variations in the adhesion and cohesion behavior in the adhesive film, they are mainly utilized for the comparative assessment of adhesives and modified surfaces of substrates. To assess the adhesion, a lap shear test can be performed by pulling bonded layers apart along the plane of the adhesion [8]. Pull-out or pull-off testing is a straightforward tensile test that gives detailed data about adhesives, such as sample elongation, single-strand breaks, and slippage within the joint [55].

Methods such as X-ray photoelectron spectroscopy (XPS), time-of-flight secondary ion mass spectrometry (ToF-SIMS), and contact angle measurements are indirect measurements of adhesion. These methods can combine direct measurements for the highly accurate investigation of adhesion. Table 2 represents other possible characterization assays related to unfavorable happening after applying the tissue adhesive.

## 5. Modification of Tissue Adhesives

In addition to their use as glues or sealants, tissue adhesives can be applied as controllable drug delivery systems. The fabrication of a tissue adhesive can be designed to deliver pre-loaded contents in the desired pattern to the target site. Local release by the tissue adhesive enables the administration of safe dosages and minimizes the drug’s adverse effects [34]. Moreover, incorporating a drug delivery system in a tissue adhesive makes the patients feel more comfortable because they do not need to take the drug regularly. Today, several active agents have been investigated using this model. Accordingly, compounds such as chemotherapeutics, antibiotics, growth factors, analgesics, and gene vectors are under evaluation for their use in tissue adhesives.

It has been proven that antibiotic concentration at the wound site can prevent infection within the lesion (e.g., of surgical wounds). The local drug concentration is far lower than dosages that are commonly used in conventional dosing methods. Thus, this approach prevents the occurrence of the high dosage adverse effect. In a study by Marone et al., antibiotics such as teicoplanin, vancomycin, cephalothin, and gentamicin have been incorporated into the thrombin component of a fibrin glue [56]. The drug containing tissue adhesive was then applied into the epidermis of the wound. The developed system exhibited antibacterial activity against clinical isolates of *Staphylococcus epidermidis*. In another experiment, Amikacin was loaded in a fibrin sealant/polyurethane mixture and then implanted subcutaneously in the anterior abdominal region of rats [57]. Based on the results, the antibiotic activity was detectable in blood samples for up to 24 h. Moreover, the maximum local concentrations of Amikacin in tissue near the glues were 210 times higher than when the drug was given systemically.

Anesthetics are another active agent that have recently been loaded into tissue adhesives. Regarding this type of active agent, Zhibo et al. developed fibrin glue adhesives containing lidocaine to alleviate breast pain after subpectoral breast augmentation surgery [58]. This approach significantly reduced pectoral pain of patients in the treatment group compared to patients who received lidocaine or fibrin glue alone. In another study, tonsillar fossae of patients were coated with fibrin glue containing lidocaine (dissolved in a thrombin solution) immediately after tonsillectomy. Patients in the treatment group returned to a normal diet after 3.7 days with no complications observed [59].

Chemotherapeutics have also been included in photocross-linkable chitosan tissue adhesives in cancer patients. Through this approach, the systematic delivery side effect of the drug could be reduced remarkably, whereas the local concentration of the drug remained at a high level at the tumor site [60]. According to this principle, antineoplastic drugs such as paclitaxel were incorporated into photocross-linkable chitosan tissue adhesives [61,62]. The results exhibited that the paclitaxel-containing hydrogel prevented the growth of subcutaneous tumors more effectively than those treated with drug-free chitosan gel or free paclitaxel injected subcutaneously at the tumor, for a minimum of 11 days.

Another ideal active molecule to deliver to wounds through adhesives are growth factors. For instance, in this approach, a matrix to promote wound healing has been fabricated by including recombinant human epidermal growth factor (rhEGF) into a photocross-linkable mixture of glycidyl methacrylated chitooligosaccharide and di-acrylated Pluronic F127 [63]. Afterwards, the wound adhesive was placed on dorsal burn wounds in rats, where epidermal differentiation was significantly promoted compared to hydrogel without rhEGF. Fibrin sealant have also been exploited to deliver nerve growth factors (NGF) into the site of sutured peripheral nerve. Compared to groups that received NGF or fibrin sealant alone, an increase in regenerated nerve fibers could be observed at the target site. Additionally, fibrin sealants have been used to deliver gliaderived neutropic factors (GDNF). This method had a more notable in vivo effect on neuron growth than using the free factor or the sealant alone [64,65].

Until now, some experiments have been performed to use the potential of wound adhesives to immobilize cells onto wet tissue surfaces. In one study, a star-shaped PEG core with DOPA end groups was introduced for cell immobilization. After fabrication of DOPA containing PEG 3D hydrogel, islet cells were incorporated into the hydrogel before the oxidization process. It was reported that this adhesive material maintained an intact interface with the supporting tissue for up to 1 year. It could also be demonstrated that encapsulated cells could remain normoglycemic for over 100 days [66].

## 6. Silk as Biomaterial

For centuries, silk has been obtained from domesticated silkworms (e.g., *B. mori*), especially for the production of high-quality textiles. An already well-established process and the beneficial properties of the material led to its biomedical and therapeutic significance being extensively studied in recent years [23]. The main component of caterpillar silk is fibroin. This is a fibrous protein secreted from specialized glands by the animal [23]. The silk filaments consist of two parallel-arranged fibroin filaments connected by a sericin layer. After a so-called “degumming” process, the sericin is separated out and the fibroin filaments are dissolved. The secondary ß-sheet structures embedded in amorphous ground substance contain glycine, alanine, and serine in particular, and the repeat sequences vary among species, causing variations in chemical and mechanical properties (Figure 1 and Figure 2). Silk is considered one of the strongest fibers found in nature (5–12 GPa with sericin, 15–17 GPa without sericin). Its stability can be explained by the molecular structure of the polymers. Large hydrophobic domains are interrupted by short hydrophilic regions, which result in a dense spatial arrangement of protein chains [23].

In addition, silk is extremely light (1.3 g/cm^3^), elastic, and has a thermal stability of up to 250 °C. For this reason, it is possible to process the material under extreme chemical and physical conditions. Minimal immunological reactions, good biocompatibility, and promotion of cell adhesion and proliferation, cell growth, and differentiation offer great potential for numerous application areas, such as biomedicine, bioengineering, materials science, and the pharmaceutical industry [67]. The aforementioned properties make silk a versatile material for the natural regeneration of damaged tissue. Furthermore, slow degradation and insolubility in common solvents such as water, dilute bases, acids, and alcohol play a crucial role. Silk proteins can be processed into numerous delivery forms, such as gels, membranes, nanofibers, meshes, particles, and foams, making them particularly interesting for drug delivery systems and tissue engineering [68]. Chemical and mechanical modifications of the nanostructure can for instance influence degradation time, solubility, therapeutic effects, morphology, and functionality [23].

Furthermore, it is possible to combine silk proteins with other natural and synthetic polymers to improve the properties of the biomedical product [23]. One example is tannic acid, a polyphenol that can be extracted from plants. Since natural polymers often lack sufficient intrinsic antimicrobial activity, cross-linking seems quite reasonable. The substance is said to have antibacterial, antiviral, anti-inflammatory, and anticarcinogenic properties [69]. Furthermore, it exhibits antioxidant and hemostatic activity and helps neutralize free radicals, which plays a crucial role in a variety of diseases. As a combination partner for numerous biomedical products, tannic acid is a promising option due to its high viscosity, elasticity, and straightforward processing [12,67,68].

## 7. Silk-Based Adhesives

In addition to the established fully synthetic and semi-synthetic adhesives, more and more biologically based formulations are finding their way into research, clinical trials, and clinical applications. Silk fibroin-based adhesives offer one such biological alternative. As a naturally degradable biomaterial, silk fibroin represents an interesting alternative to conventional medical adhesives.

The degradation time can be individually adapted to the specific needs of the application area, so that wounds have sufficient time to heal. Since silk proteins cannot be dissolved in common solvents, the tissue adhesive could also be used internally. Unlike mussel proteins, which must be produced synthetically due to the unproductive extraction process, silk can be extracted directly from the animal and is ready for immediate processing, making it be easier to maintain natural functionality. Limitations include the limited data available on silk protein-based tissue adhesives. However, silk is already being investigated in many areas of regenerative medicine [23]. Silk Fibroin methacryloyl can be employed as an adhesive, a sealant, and a hemostatic agent. Kim et al. used this versatile medical glue to develop an effective sealant. They reported that methacrylated silk fibroin sealant is entirely biocompatible and also has rapid cross-linking [70]. The results of mechanical tests and ex vivo aorta pressure tests demonstrated that this sealant also has superior physical properties. Moreover, the outcomes of in vivo biological tests on the skin, liver, and blood vessels of rats represented excellent adhesive and hemostatic ability and faster wound healing than other cemeterial product. They concluded that this photocurable silk fibroin glue in an appropriate substance to apply as an adhesive, sealant, and hemostatic agent.

Another approach was developed by Gao et al., who developed a novel tissue adhesive in powder form based on silk fibroin and tannic acid. The powder mixture is simply mixed with water and can, thus, be used directly. One disadvantage, however, is that the mixture must be stored at −20 °C. The authors describe good adhesion strength under aqueous conditions, good elasticity, and biocompatibility. The drying time of several hours until fully formed stability is to be criticized here. Furthermore, a comparison with cyanoacrylates and fibrin adhesives would have been interesting [13].

Shin et al. developed a DNA-tannic acid hybrid gel that could also be prepared by simple mixing. The material proved to be extremely stable (>10 kPa), adhesive, and stretchable in mechanical tests. In addition, good hemostatic efficacy was demonstrated in a mouse model [14]. One year later, a promising mucoadhesive glue was developed by combining tannic acid and PEG, which was tested in vivo for the first time. Adhesive properties were further investigated in the esophageal mucosa of a rat, where the adhesive adhered for several hours [71].

In an experimental study by Burke et al. [72], silk fibroins inspired by mussels were modified with catecholamine groups to improve adhesiveness. Furthermore, it was possible to control solubility in water by addition of a PEG side chain. The ability to form secondary ß-sheet structures remained unaffected despite the modifications, so that stability could be maintained. The authors hypothesize that despite the increased water solubility, the hydrophobic core of the fibroins results in a low swelling index. However, the complex production process and, similar to Gao et al., an extremely long drying time of several days are problematic. Serban et al. [15] modified already-existing PEG-based tissue adhesives with silk fibroin. The original high swelling index could be reduced in this way and the degradation time was extended.

In another study, Liu et al. [73], fabricated a microstructured silk fibroin-based adhesive for flexible skin sensors. They demonstrated that their engineered structure was highly comfortable, adjustable, highly conformal, and biocompatible. In addition, they observed that this adhesive shows tunable adhesive properties on the skin surface and establishes a consistent bonding force on the skin surface, even under humid or wet conditions, and can be easily separated from the skin without causing pain. Due to these ideal characteristics, Liu et al. suggested that their functional system is greatly promising for different epidermal electronic sensors in the field of personalized healthcare [73].

For a long time, sericin was considered a waste product of the silk industry, extracted during the degumming process due to its potentially strong immunological properties [23]. It is now known that pure sericin is harmless; on the contrary, it seems to be a potential candidate for the development of a silk biomaterial due to its strong bioactivity, good cytocompatibility, low immunogenicity, and promotion of keratinocytes and fibroblasts [74]. The natural adhesive could further be considered as a base for a biomimetic tissue adhesive. In 2019, Dong et al. [75] identified a new sericin gene that is thought to be crucial for the larvae’s ability to adhere to surfaces. Silk moths are capable of producing silk throughout their larval stage. This involves the production of silk species that are not intended for cocoon production. This non-cocoon silk contains numerous proteins whose functions remain unclear to date.

Silk fibroin-based materials emerged in previous studies as potential tissue adhesives that even adhere to moist surfaces [6,72]. Still, the biochemical interactions, which are responsible for the adhesiveness of silk proteins, are not yet fully investigated.

## 8. Common Mechanisms of Silk Fibroin Adhesion

The basic mechanisms of adhesion in tissue adhesives can be classified as mechanical coupling, molecular bonding, and thermodynamic adhesion [76]. Among these mechanisms, molecular bonding is the most commonly accepted mechanism. As for mechanical coupling, two substances can be strongly bound to each other by mechanical anchoring and hooking with nubby nodules. In addition, adhesion can be established mechanically by simple roughening, which increases the contact area [77]. It should be noted that such mechanical interlocking is also plausible at the molecular level and can be varied depending on the viscosity of the substances [78]. On the other hand, molecular bonding to surfaces has been described as a key factor for improved adhesion, making the interface between tissue and adhesive flexible [78]. Generally, interatomic and/or intermolecular forces develop between the molecules at the surface of the tissue and the molecules of the adhesive, which are actually a combination of hydrogen bonding, capillary forces, van der Waals forces, static electric force, and covalent bonds [76].

### 8.1. Hydrogen Bonding

From a chemical point of view, a hydrogen bond can be established between a hydrogen atom and nitrogen or oxygen as two electronegative atoms. Depending on the molecules and their statements (gas or a solution), the strengths of the hydrogen bond can vary from 5 kJ/mole to hundreds of kJ/mole [79]. Hydrogen bonds have been extensively utilized for developing self-healing polymers by incorporating strong and reversible noncovalent hydrogen bonding moieties into the polymer structure [80]. Occasionally, some substances can be added to polymers to provide more effective hydrogen bonds in a polymer structure. Tannic acid (TA) is a natural polyphenol with multiple hydroxyl groups, which provide rich hydrogen bonds in the polymer structure. In an experiment, Xing et al. [81] developed a hybrid GelMA (Gelatin-Methacryloyl) hydrogel by incorporating tannic acid for wound closure. They observed that a high concentration of TA can enhance the mechanical property of hydrogen and increase the adhesion ability. According to literature, silk fibroin, owing to having carboxyl groups, has the capacity for hydrogen bond formation to form α-helices, β-sheet crystals, and random coils. This biopolymer, through the formation of hydrogen bonds, can interact with keratin [82], chitosan [83], hyaluronic acid [83], polyvinyl alcohol, and polyacrylonitrile-co-methyl acrylate [84]. Hence, it is believed that coupling silk with biocompatible macromolecules via hydrogen bonding is promising for developing novel biomimetic materials with controlled properties and improved biological compatibility [84].

### 8.2. Static-Electric Force

Electrostatic adhesion works based on the simple principle of attraction between the adhesive and the substrate (i.e., tissue) where there are free electronic or ionic charge groups on the surfaces [85]. Electrostatic forces have a predominant effect on particle movement, and are also stronger than van der Waals forces [86]. As aforementioned, silk fibroin is made of carboxyl and amine groups containing repetitive units. These functional groups, in an appropriate condition, can be changed to ionic species. Some parameters, such as pH of the medium and the presence of some metal ions, can affect the charge status and conformation of silk fibroin [87]. Silk fibroin is enriched with carboxylic acid (COOH) as functional groups on serine residues. In order to increase the functionality of carboxyl groups, their structure should be transformed into COO^-^ as an anionic residue. For this purpose, H^+^ should be removed from the carboxyl group by increasing the pH of the solution using aqueous basic solutions such as NaOH. The reactive carboxyl group can be electrostatically bound to positively charged substrates and establish the adhesion. On the other hand, NH^2^ groups in amino acids are known as strong nucleophiles. This means that in acidic solutions, NH^2^ groups tend to absorb H^+^ and convert to NH^3+^. Therefore, these positively charged groups can be electrostatically attached to negatively charged substrates [88]. For activation of both carboxy and amino groups, the required pH variation for ionization directly depends on pK of dominant aminoamides in protein [89].

### 8.3. Van der Waals Forces

Van der Waals forces arise from a temporary dipole moment produced by the instantaneous positions of electrons in any polar or nonpolar hydrophobic molecules. They are the weakest among all intermolecular forces, but they become significant when a large number of particles are involved at a suitable (nanoscale) distance [90]. Additionally, this force becomes dominant in the case of hydrophobic surfaces. Silk Fibroin is a natural biopolymer consisting of aminoamide units. These amino acids contain primary amine and carboxyl groups that can participate in van der Waals Forces. The most abundant aminoamides in silk fibroin are Alanine (30.3%) and Glycine (45.9%), which are nonpolar and show the potential of establishing van der Waals forces in the presence of other nonpolar side chains to form α-helices, β-sheet crystals, and random coils [91].

### 8.4. Capillary Forces

Capillary force exists everywhere under natural and wet conditions. So-called meniscus force can be caused by a liquid meniscus around the contact areas of two lyophilic solid surfaces or capillary bridges of one liquid in another immiscible liquid [18]. Capillary forces significantly commit to the adhesion of biological and artificial micro-and nanoscale objects. In a study conducted by Lin et al. [92], it has been recommended that capillary forces of nanofibrils may contribute to the strong adhesion exhibited by some creatures such as abalone under humidity. Additionally, they suggested that this phenomenon can inspire future synthetic nanofibril attachment devices. Thus, this broadens its application including the use of capillary forces in wet environments. In the case of silk fibroin, Augusto Márquez fabricated a nanoporous silk fibroin film and exhibited that this natural polymer represents capillary activity. This research team used this capacity of silk fibroin to filter blood cells [93]. They suggested that the porosity together with its dual hydrophobic (by the presence of β-sheet domains) and hydrophilic nature (due to α-helix regions) were responsible for the capillary force to the silk fibroin films. Similarly, Sagnella et al. developed a biopolymer silk fibroin film using microfluidic patterns [94]. In this study, they declared that capillary force and surface tension of silk fibroin effectively participate in the self-assembly of polypeptide predefined substrates leading to silk fibroin film fabrication.

### 8.5. Chemical Bonds

Chemical bonds established at an interface can play an important role in adhesion. Chemical bonds, including covalent and ionic bonds, represent typical strength at the range of 10–1000 kJ/mol, whereas van der Waals interactions and hydrogen bonds present with less than 50 kJ/mol. The functional reactive moieties that can be exploited in adhesives using chemical bonds are alkyne, alkene, acetal, orthoester, imidoester, carbamate, thiocarbamate, carbamide, aldimine and ketimine, N-acylurea and O-acylisourea, maleimide, disulfide, epoxide, aziridine, carbodiimide, episulfide, ketene, carboxylate, phosphate, alkyl halides, aryl halides, and alkyl sulfonate esters [52]. It should be taken into account that all of these moieties cannot be used in real applications because they are potentially dangerous and can cause infection risks and tissue necrosis [95]. Two famous groups used in commercial adhesives are methacrylates for photopolymerization and NHS (N-hydroxysuccinimide)-esters or aldehydes for spontaneous cross-linking with amines [96].

Recently, NHS-ester chemistry has been commonly used in recent adhesives research due to its low toxicity and high reactivity toward amines, which are natively present in the tissue. Interestingly, silk fibroin exhibits the potential of reaction with NHS-esters to form tissue adhesives. Indeed, the NHS chemical reaction accompanies EDC (carbodiimide) to fabricate cross-link bonds between carboxyl and amine groups. Accordingly, EDC reacts with carboxylic acid groups in silk fibroin amino acids to form an active O-acylisourea intermediate that can be easily displaced by nucleophilic attack from primary amino groups in protein structure. The primary amine forms an amide bond with the original carboxyl group, and a non-toxic EDC by-product is released as a soluble urea derivative. This procedure can also be used for modification of substrate to increase its adhesion. For instance, Yin and colleagues used EDC-NHS reaction to incorporate dopamine hydrochloride into silk fibroin structure to promote its adhesion properties [51].

## 9. Conclusions and Future Outlook

In this brief review, the different tissue adhesives were compared and discussed (see Table 3). The approach of a silk-based tissue adhesive was considered in more detail.

The tendency to develop new biomimetic products is increasing with the growing demands of medicine. Mussel proteins are considered the most prominent representatives in bioadhesive research. Immunological reactions are hardly observed and the adhesive effect is satisfactory under dry conditions. In humid environments, however, the binding ability strongly decreases. In addition, the long drying and degradation time is another main disadvantage of these materials. Since none of the mentioned compounds meet the ideal conception of a capping agent, further research and studies are required. A tissue adhesive based on silk proteins is an innovative approach to get closer to creating an optimal adhesive. Thus far, hardly any immune reactions have been observed for silk proteins or pure sericin, both in vitro and in vivo. The antimicrobial effect does not seem to delay wound healing, but rather to promote it. Cell adhesion, proliferation, differentiation, and growth are also favored. By modifying the morphology, it is possible to integrate drugs and, thus, influence functionality. Furthermore, silk proteins can be combined with other substances, such as tannic acid or PEG. Silk fibroin is one of the strongest natural fibers known to man, extremely stable, and elastic and heat resistant, which allows it to be processed under extreme conditions. In addition to its excellent biocompatibility, the material is also characterized by an individually adjustable degradation time.

In addition to conventional medical adhesives, silk fibroin as a natural absorbable biomaterial represents an interesting alternative to conventional medical adhesives. Its fibroin solution, which subsequently serves as the starting material for further transformation into an absorbable adhesive, is efficiently produced by a gentle dissolving process. In addition to excellent mechanical properties, this adhesive also offers outstanding biocompatibility, and thus, clearly stands out from conventional adhesives. The incorporation of antimicrobial agents can already be carried out in the initial solution, thus ensuring homogeneous distribution of the agents within the adhesive. A silk matrix as a “release system” offers the possibility of complete degradation over a period of at least 21 days and guarantees uniform release of the active ingredients [97,98].

Within the scope of the studies, further fields of application for silk-based materials have already been explored. In particular, the field of wound pads and tamponades has proven to be a promising area in the future. Furthermore, due to its natural properties, the material can be excellently applied in the field of reconstructive and aesthetic surgery. The field of maxillofacial surgery in particular offers a wide range of indications. An example from the field of maxillofacial surgery could be the lining of the implant neck using a biodegradable tissue adhesive. This reduces the soft tissue gap to the implant and minimizes colonization by pathogenic germs through an optional and risk-adapted antimicrobial loading with delayed release of active substances. This would prevent the spread of pathogenic germs and the formation of biofilms and, in addition to preventing periimplantitis, would significantly reduce the need for suture material, patient suffering, and follow-up care. Initial investigations have already been launched in this regard.

Silk is already being used successfully in a wide range of applications. The production of bone replacement materials, the generation of artificial corneas, the regeneration of nerves and tendons, or the development of drug delivery systems represent some promising examples [23]. Conventional tissue reunification methods can induce mechanical stress and foci of infection. Therefore, it is useful for the advancement of reconstructive surgery to explore and develop alternatives. Much research is currently taking place in the field of bioadhesion. Since no material has yet met the criteria of an ideal tissue adhesive, there is still room for improvement. One difficulty is finding a product that stands out for its particular properties [5]. An improved understanding of the interplay of environmental and chemical factors will open up multiple opportunities in the field of wound closure materials [45]. Silk can be integrated into the human organism to unite tissues with minimal immunological reactions while promoting the healing process. In particular, the combination with other polymers, such as tannic acid or PEG, can further optimize the already-excellent properties of silk proteins. Their good adhesiveness, even under humid conditions, a broad availability and the diverse processing possibilities of silk proteins as a basis for a novel, biomimetic tissue adhesive offer a promising look into the future. However, many years of research will still be necessary before a product can be integrated into everyday surgical practice [12].

## Figures and Tables

**Figure 1 ijms-23-07687-f001:**
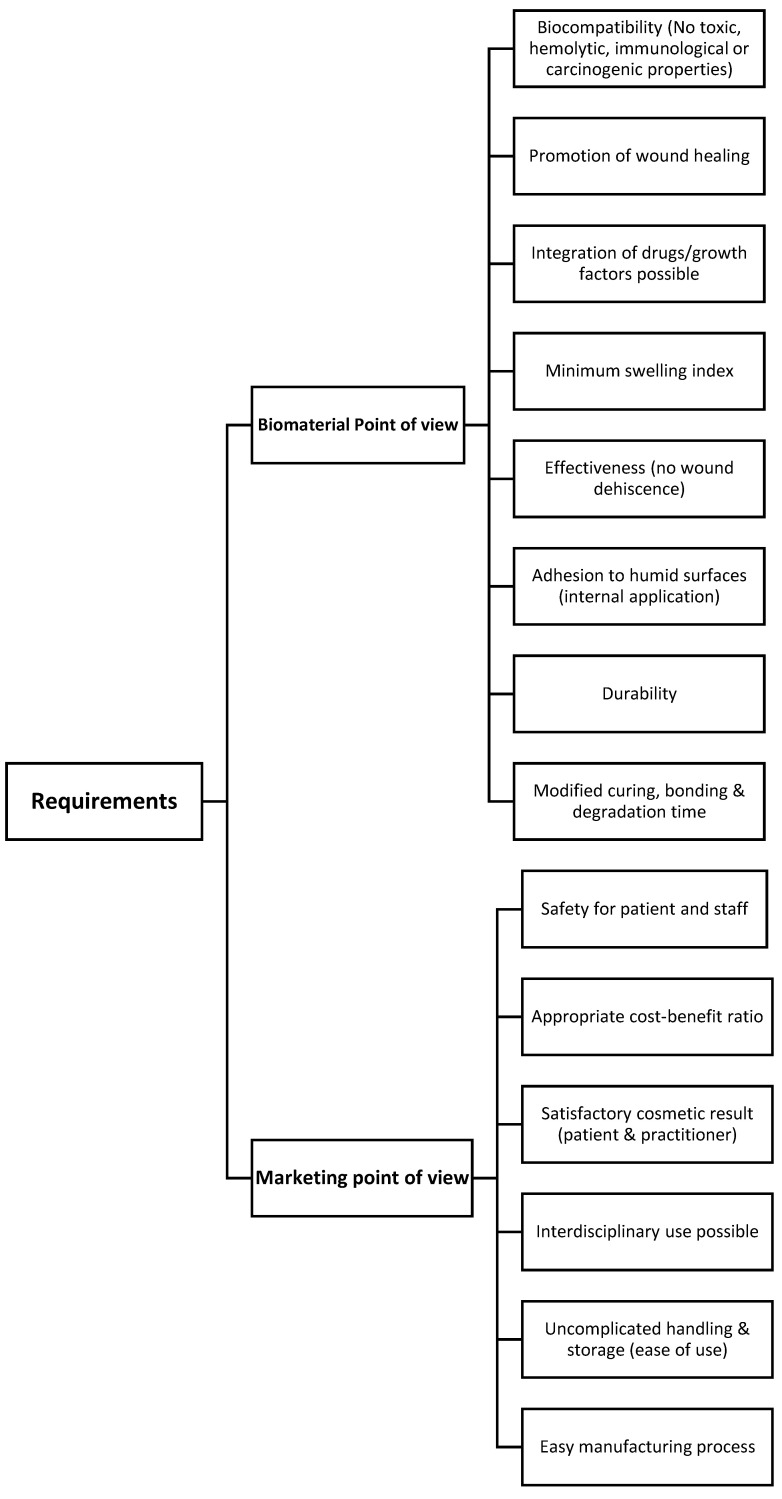
Requirements for an ideal adhesive.

**Figure 2 ijms-23-07687-f002:**
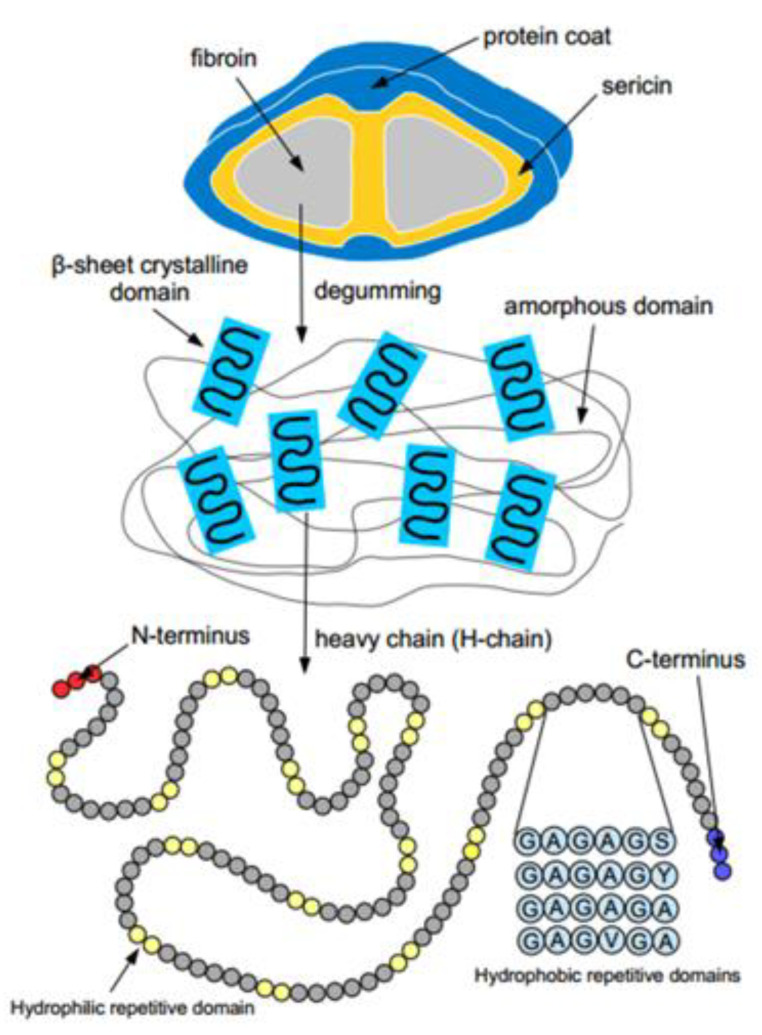
Schematic representation of silk fibers from silk spinners [23].

**Table 1 ijms-23-07687-t001:** Some specific characteristics of adhesives regarding tissue types.

Tissue Types	Adhesive Characteristic
Dura	Must maintain stability during regeneration Regeneration time is up to 1 month
Eye	Must be flexible Elastic modulus ~120 kPa
Lung	Must maintain adhesion in a dynamic environment (lung expansion) Must be elastic (elastic modulus ~5–30 kPa)
Vasculature	Must maintain its properties in wet as well as the dynamic environment (pulsatile) Elastic modulus ~0.1–1 GPa
Skin	Must be flexible Should maintain adhesion in a dynamic environment (tension) Elastic modulus ~300 kPa (epidermis)
Gastrointestinal tract	Must be elastic Material stability should be maintained during regeneration time (up to 1 month) Elastic modulus ~60 kPa

**Table 2 ijms-23-07687-t002:** Exemplary characterization assays related to unfavorable outcome.

Unfavorable Events	Characterization Methods
Unusual Inflammation Responses	Histology Proteomics FTIR and NMR spectroscopy Flow cytometry Gene expression analysis Immune phenotyping
Mechanical Failure	Mechanical tests Gravimetric analysis In vivo image tracking Histology
Swelling and Inappropriate Degradation	Gravimetric analysis In situ imaging Histology FTIR and NMR spectroscopy

**Table 3 ijms-23-07687-t003:** Comparison of the represented adhesive technologies based on its properties meeting the requirements.

Requirements	Silk-Based Adhesives	Synthetic Polymers	Polysaccharide Based Adhesives	Protein-Based Adhesives	Biomimetic Adhesives
Biocompatibility (no toxic, immunological, or carcinogenic properties)	+	+	+	+	+
Promotion of wound healing	+		+	+	
Safety for patients and staff (no undesirable side effects (e.g., allergic, dermatological, respiratory))	+	−	+	−	+
Possibility of drugs/growth factors integration	+			+	
Appropriate cost–benefit ratio	+	−	−		−
Minimum swelling index	+	−			
Efficiency (no wound dehiscence)	+	+		+	−
Satisfactory cosmetic result (patients and practitioner)	+			+	
Interdisciplinary use possible	+	+	+	+	+
Adhesion to humid surfaces (internal application)	+	+	−	+	−
Uncomplicated handling and storage (ease of use)	+	−		−	−
Durability	+				
Modified curing, bonding, and degradation time	+	+		+	
Uncomplicated manufacturing process	+	+	−	−	−

## Data Availability

No new data were created or analyzed in this study. Data sharing is not applicable to this article.

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
