# Peer review of "Tissue Adhesives in Reconstructive and Aesthetic Surgery—Application of Silk Fibroin-Based Biomaterials"

_ijms, 2022, doi:10.3390/ijms23147687_

Round 1

Reviewer 1 Report

This manuscript systematically reviewed the Tissue Adhesives, including the basic principle, the categories, evaluation methods, and modifications. The silk-based adhesives are specifically discussed. The contents comprehensively cover various ranges of tissue adhesives. I suggest adding more quantitative information about various tissue adhesives, which may help readers to establish a better understanding of various kinds of materials.  Besides, there are some format problems, Line107, incomplete bracket; Line 320-325, unnecessary underline. When these issues are well-considered, I'd like to recommend the publication in the present Journal.

Author Response

We appreciate your insightful suggestion and agree that it would be useful to include the quantitative data in the manuscript to demonstrate and compare the adhesion of different adhesives and silk-based adhesives. It is undeniable that these quantitative data add more value to our work. However, we would like to point out that according to the copy and write obligations, we need to get permission from the authors of other related papers to insert the quantitative data in the form of figures or diagrams. As you know, this procedure will cost us a lot of time. Therefore, we preferred to include only qualitative information. Regarding the format problems, we have met these matters as you can consider them in the manuscript and bellow in the manuscript:

Other Comments and Responses:

Comment: Line 107: incomplete bracket;

Response: The bracket has been completed. Also, the space between paragraphs has been omitted to give the reader a better sense.

Comment: Line 320-325: unnecessary underline.

Response: Thank you for your precise comment. The underlining has been removed from the manuscript. Please note, however, that in order to address other comments, we had to move (replace) some paragraphs in the manuscript. Therefore, the line numbers have been changed as follows:

Line 320-325 to Line 220-222

Please see the attachements

Reviewer 2 Report

This article aims to discuss the most commonly used bioadhesives and the potential of silk fibroin as a natural adhesive. This is a novel and interesting subject, but unfortunately I feel that this manuscript is more like a simple splicing of two parts rather than an organic whole. For example, the adhesive mechanism classification section can introduce more silk fibroin-based adhesives related to this adhesive mechanism. Alternatively, the introduction of silk fibroin-based adhesives can be classified and introduced corresponding to the previous "Principals of Tissue adhesives", "Common Mechanisms of Adhesion", etc.

Author Response

We think you have emphasized an important point. To address your valuable comment, we are changing the structure of the review. Sections 3, 4, and 5 have been moved (replaced) to make them more meaningful. As you can see, we first included the general content about bioadhesives and then inserted the topics related to silk-based adhesives, application, adhesion mechanism, etc... .

Accordingly, the order of sections after revision:

1- Introduction

2- Principals of tissue adhesives

3- Categories of tissue adhesives

4- Evaluation methods of tissue adhesives

5- Modification of tissue adhesives, As general parts and following section as silk-based adhesives specific content:

6-Silk biomaterials

7- Silk based adhesives

8- Common mechanisms of silk fibroin adhesion

9- Conclusion and future outlook 

See the attachments

Round 2

Reviewer 2 Report

I am very grateful to the author for answering my question carefully and meticulously, and making appropriate revisions. Therefore, I think this manuscript deserves acceptance.